# Investigating the impact of HIV on patients with first episode psychosis: a study protocol for a longitudinal cohort study

Usha Chhagan ,[1] Vuyokazi Ntlantsana ,[1] Andrew Tomita,[2,3] Thirusha Naidu ,[4] Bonginkosi Chiliza ,[1] Saeeda Paruk[1]

[1]Department of Psychiatry, School of Clinical Medicine, College of Health Sciences, University of KwaZulu-Natal, Durban, South Africa
[2]KwaZulu-Natal Research Innovation and Sequencing Platform (KRISP), College of Health Sciences, University of KwaZulu-Natal, Durban, South Africa
[3]Centre for Rural Health, School of Nursing and Public Health, College of Health Sciences, University of KwaZulu-Natal, Durban, South Africa
[4]Department of Behavioural Medicine, College of Health Sciences, University of KwaZulu- Natal, Durban, South Africa

**Correspondence to**
Dr Usha Chhagan;
uchhagan@hotmail.com

## ABSTRACT

**Introduction** South Africa (SA) has a high HIV prevalence and limited mental healthcare resources. Neuropsychiatric complications such as psychosis onset in people living with HIV (PLWHIV) remains poorly understood. The study aims to compare the socio-demographic, clinical, substance use, cognitive and trauma profile of PLWHIV presenting with first episode psychosis (FEP) to those with the condition but without HIV.

**Methods and analysis** This study will compare presentation, course, and outcome of a cohort of PLWHIV and FEP with a control group recruited over a 3-year period. We will prospectively test the hypothesis that the 2 groups are socio-demographically, clinically and cognitively distinct at illness presentation, with higher trauma burden and poorer outcomes in those with the dual burden of HIV and FEP. FEP participants, confirmed by a structured neuropsychiatric interview, will have their socio-demographic, psychosis, mood, motor, trauma and substance use variables assessed. A neuropsychological battery will be completed to assess cognition, while quality of life, psychotic symptoms and HIV markers will be measured at 3, 6 and 12 months.

**Ethics and dissemination** The study protocol has been reviewed and ethics approval obtained from the Biomedical Research Ethics Committee (BC 571/18) of the University of KwaZulu-Natal. The results from this investigation will be actively disseminated through peer-reviewed journal publications and conference presentations.

## INTRODUCTION

Psychiatric illness in the presence of HIV and AIDS has been linked to negative health behaviours and poorer clinical outcomes.[1] The prevalence of HIV among those with serious mental illness (SMI) ranges from 1% to 24%.[2] Left untreated, mental disorders in people living with HIV (PLWHIV) result in poorer quality of life, greater interpersonal difficulties, increased suicide risk and poor adherence to antiretroviral therapy.

Psychotic disorders are associated with a significant burden of disease, with affected persons often relying on limited public sector mental health resources in low-and-middle-income countries (LMICs).[3] There is a lack of research on HIV and psychosis in LMICs, particularly sub-Saharan Africa (SSA), despite this region accounting for more than two-thirds of the world's HIV-positive population.[4] While research from high-income countries has demonstrated that there is a bi-directional relationship between HIV/AIDS and mental illness, there may be differences in the way this relationship is expressed in SSA.[5] There is much needed cross-cultural insight into the relationship between HIV/AIDS and psychosis. Culture may influence the patients' perception and experience of psychosis as well as the clinical presentation. This highlights the need to explore the impact of culture on psychosis.[6] Studies have also shown variations in the socio-demographic and clinical profile of those with dual burden of HIV and first episode psychosis (FEP) compared with those with psychosis only.[7]

### Strengths and limitations of this study

► This study will be among the first to examine the interacting roles of HIV, substance use and trauma in a first episode psychosis (FEP) study in Africa.
► The study may provide valuable insight into the long-term impact of HIV infection on psychosis presentation which is poorly understood in low-and-middle-income countries.
► Findings from this study will contribute towards baseline data to guide future studies of patients with FEP.
► The study is hospitals-based, and this may introduce sample bias and limit generalisability to community samples.
► Randomised controlled trial design is not possible given that study monitors disease progression.

BMJ

## Socio-demographic profile of PLWHIV and first episode psychosis

There is limited literature on the association of HIV and FEP, with previous studies mainly describing HIV prevalence rates or clinical profiles.[8] A Ugandan study by Lundberg and colleagues[9] found that HIV-positive patients with mental illness were more likely to be women and older (40–49 years). A KwaZulu-Natal (KZN) Province, SA, study of patients with FEP reported a higher HIV prevalence in less educated individuals; however, this was a small sample.[8] In a more recent chart review of patients with psychosis admitted to a psychiatric hospital in SA, HIV-infected patients with psychotic disorders were more likely to be women (74.0%), younger than 50 years (94.0%) and less likely to have secondary education than HIV-negative patients with psychotic disorders (56.0% vs 72.0%).[10]

## PLWHIV and psychosis symptomatology

In a review of the literature, grandiose, persecutory and somatic delusions were the most common psychotic symptoms, followed by hallucinations and mood symptoms in PLWHIV with FEP.[11] An Italian study found that HIV infection increased the severity of symptoms (more frequently paranoid delusions), was associated with a greater impairment in attention and concentration, and had decreased depressive symptoms.[7] In addition, PLWHIV and psychosis are more likely to be diagnosed as having a psychosis secondary to another medical condition than a primary psychotic disorder, and have more comorbid medical disorders and increased haematological test abnormalities.[10] A study in Gauteng province, SA, found that among PLWHIV presenting to psychiatry services, 23 individuals were diagnosed with psychosis due to another medical condition while only two were diagnosed with schizophrenia.[12] However, there is extremely limited literature on HIV and psychosis symptomatology, particularly from Africa.[8 13] This suggests a hiatus in the current literature on understanding the nature and severity of psychotic symptoms in PLWHIV, particularly in the African context. There is a need for systematic studies of larger cohorts with standardised tools to better understand how HIV may potentially influence psychosis presentation and course, as well as HIV disease outcomes.

## PLWHIV and FEP and cognition

Cognitive impairment is well established in both psychotic disorders and HIV infection, being exacerbated in psychotic patients with HIV.[13] Being HIV positive is traditionally associated with a sub-cortical cognitive impairment, the prevalence of cognitive deficits being reported to range from 4% to 99% in a systematic review of HIV research in SSA.[5]

De Ronchi et al[7] described cognitive screen deficits in a sample of 22 patients with FEP living with HIV in Italy. Participants had impaired attention and concentration on the Mini Mental State Examination (MMSE) but no other differences on cognitive impairment. A study of cognitive dysfunction among 156 HIV infected and 322 HIV non-infected patients with psychosis in Uganda assessed them with the MMSE and a neuropsychological battery. They found that PLWHIV and psychosis were more cognitively impaired than HIV-negative psychotic patients at baseline, and although there was some improvement with treatment, they remained more impaired at 6 months, suggesting that HIV worsened cognitive dysfunction in psychosis.[13] While earlier studies have suggested that psychosis and impairment in cognition were later manifestations of HIV, the Ugandan study demonstrated that both these conditions occur early, as shown by the average CD4 count of 305 cells/µL and the intermediate WHO staging of disease presentation. In addition, impairment in cognition also remained, despite an improvement in the psychotic symptoms.[13] Further research is warranted to establish the impact of HIV and psychosis on cognition, and whether HIV-related cognitive changes are associated with increased risk for psychosis or modifies psychosis onset or course.

## PLWHIV and FEP and substance use

Studies traditionally suggest an association between substance use in FEP patients and HIV infection.[1] Substance use patterns vary across the different regions of SA. This may be related to both ethnic and socio-economic differences.[14] Davis et al confirmed both the growing pattern of substance use as well as the trend of psychiatric comorbidity in KwaZulu-Natal (KZN). The rates of substance use in this province were also significantly higher than that reported in studies of the general population in SA.[14] Another study of patients with FEP in KZN, SA, reported lower rates of self-reported substance use among HIV-positive than HIV-negative patients with mental illness,[8] which was further supported by a retrospective chart review of patients with psychosis.[10] This may suggest that either substance use/misuse may not be a significant factor in HIV transmission in the KZN context, or that the study results were limited by underreporting or the small sample size. Thus, while the international literature suggests that PLWHIV and mental illness may be vulnerable to substance use, this has not been borne out in the local KZN studies and warrants further research.

## PLWHIV and FEP and trauma

Trauma, particularly early life trauma, is also recognised as an environmental risk factor for psychosis.[15] A South African study investigated the association between a history of traumatic experiences and the clinical features of FEP. Duration of untreated psychosis, age of and symptoms at onset were assessed in 54 psychotic patients, while 22% were HIV infected. Almost half of the study subjects had witnessed (49%) or directly experienced serious physical assault (45%), both events being associated with positive psychotic symptoms.[16]

The trauma experienced may influence psychosis and HIV outcomes, as found in a systematic review,

that hallucinations and delusions were more severe in people with a history of childhood trauma.[17] In addition, Leserman[18] reported on the negative association of HIV, stress and depression on the course of HIV in terms of decreasing the CD4 lymphocytes, increasing the viral load and having greater risk for premature mortality.

Exposure to violence and resultant trauma is of major public health concern. In SA, the mortality rate secondary to violence was estimated at 73/100 000.[19 20] While much research on early trauma has been conducted in developed countries, there is a need to examine the association of trauma and FEP in developing countries.[19–21]

## Course of comorbid HIV and psychosis

PLWHIV and psychosis are more likely to develop comorbid medical disorders, experience side effects of antipsychotic medication, develop metabolic syndrome secondary to ART, have longer length of hospital stay and readmission rates, and have poorer quality of life.[1 10] However, this has not been tested prospectively in the patients with FEP in the African context with the recent wide availability of ART.

## Aims

The aim of the study is to investigate trends in clinical, substance use, cognitive and trauma presentations in individuals experiencing first episode of psychosis over time, and to compare presentation, course and 12-month outcomes between those with and without HIV.

## METHODS

Patients referred to the participating psychiatry units will be screened by the respective treating doctors for eligibility for recruitment into the study. The patients with FEP must fulfil the inclusion criteria and will be referred to the designated investigator at each unit. Once written informed consent has been obtained, the protocol for data capture will be followed, as per the list outlined in figure 1 and table 1, the same protocol being followed for outpatients and inpatients. The clinical examination and psychiatric tests, which take approximately 2 hours, will be undertaken by the investigators who are all medical practitioners. Patient baseline assessments will be undertaken within 6 weeks of treatment during their outpatient visit or following admission to the respective units at baseline. A physical examination, body mass index and waist circumference measure will also be obtained by a medical practitioner. Where possible, collateral information from a next of kin, with the participant's consent, will also be sought. The follow-up visits will be done at 3, 6 and 12 months as outpatients, and the participants will be reimbursed for travel costs. Attempts will be made to coincide each of these with scheduled clinic follow-up visits and will be performed by the same initial interviewer at each psychiatry outpatient department.

The cognitive battery assessment, which takes approximately 2 hours, will be administered by a clinical psychologist, at the 3-month and 12-month visits. Translation services will also be available for patients who require isiZulu translation during the interviews.

Following collating the data for steps 1–13 in figure 1 and table 2,[22–33] we will perform steps 14 and 15 after voluntary counselling and testing. The required HIV tests, CD4 count and viral load if HIV positive, as per the protocol, will be taken by a medical practitioner at each site. HIV testing will be carried out at baseline for all participants as well as at each visit for HIV-negative participants. A patient with a positive HIV test will be referred to the HIV clinic at the site for further management. Other investigations, such as lipid profile, which are part of routine care, will be captured from the clinical records.

We are keen to explore impact of pathways of care on socio-demographic and clinical variables. Rehabilitation programmes are limited as many sites lack a multidisciplinary team to support this; however, standard of care includes medication, supportive therapy, psycho-education, referral to psychology if indicated and management of comorbid substance use.

## Study design and setting

The comparative cohort study will be in keeping with a quantitative, descriptive and longitudinal 12-month design of adult patients presenting with FEP that are either HIV infected or non-infected. The study will take place in the eThekwini District, one of 11 in KZN Province, with an estimated population of 3 442 361 people. The district has a high HIV burden, with an estimated

---

**Objective 1**

To compare HIV infected and HIV non-infected patients with FEP at psychosis onset in at baseline, 3-, 6- and 12-months regarding socio-demographic profile, clinical features, substance use patterns, traumatic life events and quality of life.

1. A structured socio-demographic and clinical questionnaire: age; gender; marital status; educational and employment levels; medical, psychiatric and trauma history.
2. Mini International Neuro-psychiatric Interview (MINI) version 7.02: to confirm DSM V diagnosis of psychotic disorder [36]
3. The Positive and Negative Syndrome Scale (PANSS) [37]
4. Physical examination: for weight, height and body mass index
5. Duration of Untreated Psychosis (DUP) and pathways to care assessed with the WHO Pathways to care questionnaire, WHO Encounter [38]
6. Patient Health Questionnaire (PHQ-9) [39]
7. Cognitive screening tools: Cognitive Assessment Tool (CAT) - rapid [40] and International HIV Dementia Scale (IHDS) [41]
8. Cognitive battery (described in table 2)
9. Extrapyramidal Symptom Rating Scale (ESRS) [42]
10. Childhood trauma questionnaire Short Form (CTQ-SF) [43]
11. Posttraumatic Stress Disorder (PTSD) checklist for DSM-V (PCL-5) – Life Events Checklist (LEC)-5 and extended criterion [44]
12. WHO Quality of Life Brief Version [45]
13. WHO Alcohol, Smoking and Substance Involvement Screening Test [46]

**Objective 2**

To describe the association of HIV infection clinical markers (CD4 count and viral load) and clinical variables of psychosis at illness presentation, 3, 6 and 12 months.

14. HIV ELISA testing for all participants
15. HIV markers- CD4 count and viral load for PLWHIV at baseline, 3, 6- and 12-month visits.

**Figure 1** List of all study procedures used in the patient assessment.

**Table 1** Objectives and measures used at each visit in HIV-infected and non-infected participants

| Domain | Measures | Source of measures/tool | Visit 1 Within first 6 weeks | Visits 2 and 3 At 3 and 6 months | Visit 4 At 12 months |
|---|---|---|---|---|---|
| Demographic factors | Socio-demographic data | Socio-demographic questionnaire | X | | |
| Mental health outcome (objective 1) | Psychosis | Psychiatric history and MINI | X | | |
| | DUP and pathways to care | WHO Encounter | X | | |
| | Severity and nature of psychosis symptoms | PANSS | X | X | X |
| | Depression | PHQ9 | X | X | X |
| | Trauma | CTQ | X | | |
| | | PCL | | X | |
| | PTSD | Clinician-administered PTSD scale for DSM-V (CAPS 5) | | X | |
| | Motor symptoms | ESRS | X | X | X |
| | HIV-associated cognitive deficit screen | CAT-Rapid and IHDS | X | X | X |
| | Social cognition | Neuropsychological battery | | X (at 3 months only) | X |
| | Substance use | WHO ASSIST | X | X | X |
| | Quality of life | WHO-QOL BREF | X | X | X |
| | Biochemical haematological and radiological tests | From hospital records: full blood count, liver function test, syphilis and hepatitis test. Cerebrospinal fluid biochemistry and neuroimaging if available | | | |
| Clinical markers of HIV (objective 2) | HIV status | Confirmatory HIV ELISA | | | |
| | HIV marker | VL if HIV positive | If applicable | If applicable | If applicable |
| | HIV marker | CD4 if HIV positive | X | | |

ASSIST, Alcohol, Smoking and Substance Involvement Screening Test; CAPS, Clinician-Administered PTSD Scale; CAT, Cognitive Assessment Tool; CD, cluster of differentiation; CTQ, Childhood Trauma Questionnaire; DUP, Duration of untreated psychosis; ESRS, Extrpyramidal Symptom Rating Scale; IHDS, International HIV Dementia Scale; MINI, Mini International Neuropsychiatric Interview; PANSS, Positive and Negative Syndrome Scale; PCL, Psttraumatic Stress Disorder Checklist; PHQ, Patient Health Questionnaire; PTSD, Posttraumatic Stress Disorder; QOL BREF, Quality of Life; VL, viral load.

650 000 people living with the virus in 2018.[34] In addition, there are only five hospitals in the municipality with psychiatric services managed by a psychiatrist, at which the study will be conducted. The hospitals selected are the four general hospitals with specialist psychiatric services situated in eThekwini: Addington, King Edward VIII, Prince Mshiyeni and RK Khan Hospitals, and at King Dinuzulu Hospital, which serves as the psychiatric hospital for the area. All hospitals have inpatient and outpatient psychiatric services and receive referrals from within the district as well as from urban and rural areas as far south as the Eastern Cape Province and northern KZN.

**Eligibility criteria**
**Inclusion criteria**
Male and female patients will be included if they are inpatients or outpatients, aged 18–45 years, have a first presentation to mental healthcare services for FEP meeting Diagnostic and Statistical Manual (DSM)-V criteria[35] (American Psychiatric Association (APA) 2013) for a psychotic disorder, are neuroleptic naive or minimally treated (maximum 6 weeks of psychotropic treatment in this first episode), and provide consent to participate.

**Exclusion criteria**
Patients will be excluded if they have been prescribed antipsychotics in the past or have had previous psychotic episode, an intellectual disability (assessed clinically or by chart review) or other significant general medical conditions that may be associated with FEP, for example, epilepsy or syphilis, have less than 7 years of formal schooling and refuse consent.

| Table 2 | Cognitive test battery and domains to be tested |
|---|---|
| **Domains** | **Test** |
| Motor skills | Grip strength[22], Pegboard[23] |
| Visuospatial functioning | WAIS-R Block Design, Digit symbol[24], and Rey-Osterrieth Rey Complex Figure (RCF)—Copy[25] |
| Verbal memory | Rey Auditory Verbal Learning Test (RAVLT)[26] |
| Non-verbal memory | RCF—Recall[25] |
| Problem solving/ reasoning | WAIS-R Language test[24] |
| Language | RAVLT[26] |
| Word generation | Controlled Word Association Test (COWAT)[27] Animal Naming (Letter and Category cues)[28] |
| Attention | WAIS-R Digit Span[24] |
| Executive functioning | Trail making[29], RCF[25], Wisconsin Card Sorting Test (WCST)[30] |
| Social cognition | Emotion perception (Faces Test)[31], Theory of mind (Eyes Test)[32], Social knowledge (Situational Feature Recognition Test) (SFRT)[33] |

COWAT, Controlled Word Association Test; RAVLT, Rey Auditory Verbal Learning Test; RCF, Rey-Osterrieth Rey Complex Figure; SFRT, Situational Feature Recognition Test; WAIS, Wechsler Adult Intelligence Scale; WCST, Wisconsin Card Sorting Test .

## Participant selection and sampling strategy

All adult inpatients or outpatients with FEP meeting the inclusion criteria will be recruited to the study over a 3-year period. The sample size calculation was based on a log-rank test group. In order to achieve an 80% power to detect an OR of 1.857 (to maximise variability), in a design with three repeated measurements having a compound symmetry covariance structure when the proportion from group 2 is 0.500, the correlation between observations on the same subject is 0.500, and the alpha (significance) level 0.05 or 5%, a sample of 130 PLWHIV and FEP and 130 HIV negative with FEP will be required. In addition, this sample size accounts for and permits a 10% dropout rate over an anticipated 36-month accrual with a minimum follow-up of 12 months.

## Data collection

The study will use a clinical interview, physical examination, several psychiatric tools and HIV-related haematological tests to measure variables. Baseline measures will be done within 6 weeks of first presentation and as soon as the participant is able to consent. Participants need to have responded to antipsychotic medication and will be assessed to have the capacity to consent by a treating and research doctor. The study (HIV+) and control groups (HIV−) will receive the same baseline assessments and follow-up at 3, 6 and 12 months, which are summarised in

table 1. The HIV-negative group will have HIV testing at each visit to re-confirm HIV status. The study procedures, with instruments,[36–46] that will be used are shown in the information in figure 1. Data will be collected manually and captured electronically.

Clinical researchers with a medical background will administer the mental health diagnostic assessment, the MINI 7.02, Psychotic Disorders version, DSM-V[36] and diagnostic interview face-to-face, and will be supported by a trained research assistant who will facilitate the self-report questionnaires. All study personnel will be trained on initiation of the study with an annual refresher training on the different tools by psychiatrists with research and clinical experience. A medical doctor will conduct the clinical assessments, collect the biological specimens (table 1), and conduct the HIV counselling and testing. We will use an HIV COBAS COMBI test to detect both HIV antibodies and p24 antigen. Standard pre-test and post-test counselling will be delivered in a culturally sensitive manner, while the CD4 count and viral loads will be measured at every visit among PLWHIV.

Participants will be given a return research appointment date and telephonic contact will be maintained between interviews. The study team (research assistant and clinician) will contact participants to remind them of appointments to limit their withdrawing from the study. For participants who do not arrive for their appointments, the research assistant will contact the patient to reschedule an appointment in the same week.

## Data management and analysis

All participants will be allocated a participant identification number (PID) code at enrolment, which will be used on samples and documents over the 12-month study period. Data will be downloaded to a secure server and kept in a password-protected system. Laboratory results will be merged into the main dataset using the unique PID.

Data will be entered into SPSS V.24 and the analyses conducted using STATA V.15. The socio-demographic and clinical characteristics of participants will be summarised using means and SD continuous variables. For categorical variables, proportion (%) will be reported. Significant associations in contingency tables (cross tabulations) will be assessed using the standard Pearson's $\chi^2$ test. An independent samples t-test will be used for comparing differences in continuous variables between two groups. If there are more than three groups, we will use ANOVA methods. A p value less than 0.05 will be regarded as statistically significant. In addition, we may also apply growth mixture modelling to observe whether subsets of individuals follow distinct trajectories over time.

## Patient and public involvement

No patients or members of the public were involved in the design, analysis or reporting of this current investigation.

## Ethics and dissemination

Ethical approval to conduct the study was obtained from the Biomedical Research Ethics Committee of the University of KwaZulu-Natal (UKZN). The study will be conducted in accordance with South African Department of Health Research Ethics guidelines (2015) as well as the UKZN policy on Research Ethics. Recruitment of study participants will commence as full ethics approval was received from Biomedical Research Ethics Committee. No other independent ethics review was required.

All participants fulfilling the inclusion criteria will be selected for the study, from whom written, informed consent will be obtained. All potential participants will have the opportunity to participate, which will be made known to them in the information sheet. Where participants are unable to consent independently, adequate steps will be taken to ensure that legally acceptable proxy consent is obtained. All participants testing HIV positive at any point in the study will be referred for ART and support.

The results from this investigation will be actively disseminated through peer-reviewed journal publications and conference presentations. On completion of the peer-review process, we will provide feedback to clinicians at Department of Health for further discussion about enhanced treatment algorithm.

## DISCUSSION

### Potential risks

There will be minimal risk or discomfort to participants, with one test requiring bloods to be taken. The only active involvement of the participants is for them to provide verbal responses to assessment interviews and scales used for data collection. We acknowledge the potential risk of psychological distress to patients, as they may be directly affected by sensitive questions, for example, relating to trauma during the administration of the psychiatric scales. The study has a distress plan to support such patients, which entails referral to treating team if there is concern about relapse or danger to self or others. The sites have specialist psychiatric services, both inpatient and outpatient, with a team available to attend to any adverse event that may arise during the interview process.

### Methodological challenges and study limitations

Several limitations may affect the study, including that it is hospital-based, which may introduce sample bias and limit the generalisability to community samples. Another limitation is the use of tools that have limited validity in the local South African context, as they have not been used previously. Language and cultural barriers in communication will be addressed by an available isiZulu translator. A further limitation of the observational nature of the study is the attendant lack of standardisation of treatment, for which will follow the Essential Drug List (EDL) for hospital level adults (Department of Health, 2015). Although patients will follow the same algorithm, they may be maintained on different medication, depending on the individual patient treatment response. The study will, however, provide an adequate sample representing the clinical profile of patients with FEP and PLWHIV in a resource-constrained setting.

## Study progress and challenges

A challenge we anticipate is the loss of participant follow-up, which is generally a common failing in managing psychiatric patients, particularly in LMIC settings with challenging psychosocial circumstances. The 3-month, 6-month and 12-month follow-up appointments may remain difficult to sustain, despite supporting participants with travel costs for visits. Maintaining follow-up visits is a challenge we anticipated and for which we have a contingency strategy, this being that a research assistant will have regular communication with the recruited participants, confirming appointment dates, sending reminders and facilitating travel arrangements. Participants seroconverting to HIV positive status during the study will be referred to antiretroviral services for care but continue follow-up assessments, noting the change in their HIV status.

## Study significance

We postulate that the course and outcome of HIV psychosis differs from that in an HIV-negative individual, and that this study may provide valuable insight into the impact of HIV infection on psychosis presentation and outcomes. Longitudinal comparative cohort studies of this nature are limited and have not been conducted locally in one of the highest HIV prevalence settings in Africa, hence the data will serve to introduce new knowledge on the topic, as well as form the baseline for future research and treatment. A clearer understanding of the association between HIV and psychosis may provide better insight into HIV as a neurotropic virus. The findings of this study will contribute towards improving the care and management of patients with FEP and HIV.

WHO reports that in 2020, psychiatric conditions are ranked the second leading cause of global disease burden after ischaemic heart disease.[47] In the South African context, this is even more pronounced by the additional burden of HIV infection and AIDS. With an approximate overall HIV prevalence rate of 11.2% (6.19 million) and an estimated 16.6% of the population (adults aged 15–49 years) being HIV positive, SA is in urgent need of interventions to address the related adverse consequences. Psychotic disorders are associated with a significant burden of disease and often use the limited mental health resources. Understanding the association between HIV and psychosis in the era of antiretroviral treatment is thus an urgent priority. Using the HIV psychosis as a potential model for schizophrenia may also improve our understanding of its neurobiology. The new knowledge, generated through this novel study, will enable an understanding of the interplay between HIV and psychosis at a fundamental level, centred at the international epicentre of the HIV pandemic.

In addition, in SA, substance use and trauma remain critical public health challenges that need to be further explored. Thus, the study of the bidirectional associations between HIV, substance use and traumatic experiences, psychosis onset and presentation in a low-income and middle-income setting is essential to better guide the strategic use of their limited resources. Early insights into the role of substance use and trauma in HIV infection and psychosis onset could be used to delay, modify or even prevent the onset of HIV psychosis. Furthermore, the interplay of these extraneous factors in genetically predisposed individuals has not been explored. This study will allow us to begin to understand the contribution of HIV to psychosis onset, increase awareness of the need for screening and care in a highly vulnerable group, while building research capacity.

In LMIC, 11.1% of the total burden of disease are attributable to mental disorders.[21] In addition to the high prevalence and morbidity associated with mental disorders, it has been shown that cultural influences impact significantly on and contribute to the risk for mental and other health concerns in developing countries. The results of this study will contribute to the development of management (investigation and treatment) algorithms in the South African resource-constrained health departments. Reference will be made to the differences identified between the HIV positive and HIV negative groups, facilitating appropriate tailored approaches to each sub-group. The benefits of these would include policy changes and streamlined cost-effective treatment plans that will be equally beneficial to patient and the health department.

**Contributors** UC, VN, TN and SP conceived the study. UC, VN will collect data. AT performed the statistical analysis. SP and BC supervised the work undertaken. UC wrote the first draft. All authors have read and approved the final manuscript.

**Funding** This study is funded by National Research Foundation of South Africa (Grant no. 117858), South African Research Council SIR grant and a start-up grant from the Society for Biological Psychiatry Research Fund. Funding (Grant no. N/A) was also received from University of KwaZulu-Natal College of Health Sciences PhD Scholarship (Grant no. N/A). AT was funded by UK Global Challenge Research Fund (MR/T029803/1).

**Disclaimer** The opinions and findings presented are those of the authors and not a reflection of the funders.

**Competing interests** None declared.

**Patient and public involvement** Patients and/or the public were not involved in the design, or conduct, or reporting, or dissemination plans of this research.

**Patient consent for publication** Not required.

**Provenance and peer review** Not commissioned; externally peer reviewed.

**ORCID iDs**
Usha Chhagan http://orcid.org/0000-0002-4436-2025
Vuyokazi Ntlantsana http://orcid.org/0000-0002-5882-100X
Thirusha Naidu http://orcid.org/0000-0002-8154-790X
Bonginkosi Chiliza http://orcid.org/0000-0001-5417-5920

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
