## [Reviewer comments · BMJ Open]

ARTICLE DETAILS

TITLE (PROVISIONAL)	Investigating the impact of HIV on patients with first episode psychosis: a study protocol for a longitudinal cohort study
AUTHORS	Chhagan, Usha; Ntlantsana, Vuyokazi; Tomita, Andrew; Naidu, Thirusha; Chiliza, Bonginkosi; Paruk, Saeeda

VERSION 1 – REVIEW

REVIEWER	Wawrzyniak, Andrew University of Miami, Psychiatry and Behavioral Sciences
REVIEW RETURNED	09-Dec-2020

GENERAL COMMENTS	The manuscript entitled "Investigating the Impact of HIV on Patients with First Episode Psychosis: A Study Protocol for a Longitudinal Cohort Study" details a protocol designed to examine characteristics of psychiatric patients presenting with first episode psychosis and to compare those characteristics between patients living with HIV and those without HIV. Based in KwaZulu-Natal Province, South Africa, the study will longitudinally assess socio-demographic, clinical, substance use, cognitive, and trauma factors baseline, 3, 6, and 12 months. -Abstract: No edits needed; study aims and methods are clearly presented. -Introduction: As mentioned in the 2nd paragraph on p.6, differences in the relationship between HIV/AIDS and mental illness in sub-Saharan Africa compared to high-income countries is an excellent point and distinguishes this study from others. Consider expanding upon this point in the Introduction. This study will provide much needed cross-cultural insight into the relationship between HIV/AIDS and psychosis; if possible, expand upon and underscore this cross-cultural perspective. In the 4th paragraph on p.7, consider emphasizing / expanding upon the point that psychosis is more a secondary diagnosis rather than a primary diagnosis; this is quite noteworthy. In the "PLWHIV and FEP and substance use" section on p.8, similar to my earlier comment, it is worth expanding upon the point that further research is warranted in KwaZulu-Natal Province in that cultural factors may potentially impact this relationship.
--

	Consider adding overall population rates of early life trauma in South Africa on p.8 especially if rates of trauma are higher compared to other regions in the world. -Methods: Overall, excellent selection of assessments and surveys in order to comprehensively capture the factors you are hypothesizing to explain the association between HIV status and first psychotic episode. Under the "Study design and setting" section on p.10, make a subheading for Inclusion Criteria and Exclusion Criteria. In the "Participant selection and sampling strategy" section, the power calculation is statistically sound; the n = 130 in each group should successfully determine the hypothesized differences between groups. The wording is a bit awkward; consider rewording the last two sentences to: "In order to achieve an 80% power to detect an odds ratio of 1.857 (to maximise variability), in a design with three repeated measurements having a compound symmetry covariance structure when the proportion from group 2 is 0.500, the correlation between observations on the same subject is 0.500, and the alpha (significance) level 0.05 or 5%, a sample of 130 PLWHIV and FEP and 130 HIV negative with FEP will be required. In addition, this sample size accounts for and permits a 10% dropout rate over an anticipated 36-month accrual with a minimum follow-up of 12 months." For the "Data management and analysis" section on p. 12, although some of the data can be analyzed using a repeated measures ANOVA, consider using growth mixture modeling that is more robust. -Discussion: Provide some specifics of the distress plan under the "Potential risks" section on p. 13. Consider adding and expanding upon the cultural component of this study; it's an important part of this protocol and has meaningful implications. You can potentially add a bit more to the 2nd to last paragraph in the Discussion on p. 15. -Tables and Figures Tables 1 and 2 are clearly presented; no edits needed. In Figure 1, add a brief description of each study objective as part of the "Objective 1" and "Objective 2" headings.
--	---

REVIEWER	NHUNZVI, CLEMENT University of Zimbabwe, Rehabilitation
REVIEW RETURNED	29-Dec-2020

GENERAL COMMENTS	Thank you for your very important study. I have a few methodological review comments and suggestions.  1. Are there any measures in place for the qualification of the diagnosis of FEP given the low mental health literacy and stigma in the study setting? 2. What are the plans for handling those who fail to stabilize after the 6 weeks of treatment? or the prevalent relapses associated with psychosis? 3. Spell the SOP training for the involved medical personnel to minimize incomplete data sets. 4. Any acknowledgement of potential confounders given the multiplicity of healthcare pathways for mental illnesses in SSA? Also, spell out if any rehabilitation treatments like occupational therapy are part of the standard of care as they are evidenced to influence the quality of life and other target outcomes. 5. Is there a justification for only isiZulu translation given the range of local languages spoken in SA? 6. What measures are in place for aiding attention and concentration as the participants take the 2hr cognitive assessment battery? 7. How often do you plan to test for HIV in the control group, and how will the positive cases be handled? Thank you.
--

VERSION 1 – AUTHOR RESPONSE

Reviewer 1

1. Introduction: As mentioned in the 2nd paragraph on p.6, differences in the relationship between HIV/AIDS and mental illness in sub-Saharan Africa compared to high-income countries is an excellent point and distinguishes this study from others. Consider expanding upon this point in the Introduction. This study will provide much needed cross-cultural insight into the relationship between HIV/AIDS and psychosis; if possible, expand upon and underscore this cross-cultural perspective.

Response: This has been further expanded on page 4.

2. In the 4th paragraph on p.7, consider emphasizing / expanding upon the point that psychosis is more a secondary diagnosis rather than a primary diagnosis; this is quite noteworthy.

Response: Additional reference and details of regional study has been added to page 5.

3. In the "PLWHIV and FEP and substance use" section on p.8, similar to my earlier comment, it is worth expanding upon the point that further research is warranted in KwaZulu-Natal Province in that cultural factors may potentially impact this relationship.

Response: This area has been expanded upon in page 6.

4. Consider adding overall population rates of early life trauma in South Africa on p.8 especially if rates of trauma are higher compared to other regions in the world.

Response: The rates are higher. Rates for violence have been added on page 7.

5. Under the "Study design and setting" section on p.10, make a subheading for Inclusion Criteria and Exclusion Criteria.

Response: Sub-heading has been added on page 9.

6. In the "Participant selection and sampling strategy" section, the power calculation is statistically sound; the n = 130 in each group should successfully determine the hypothesized differences between groups. The wording is a bit awkward; consider rewording the last two sentences to:

"In order to achieve an 80% power to detect an odds ratio of 1.857 (to maximise variability), in a design with three repeated measurements having a compound symmetry covariance structure when the proportion from group 2 is 0.500, the correlation between observations on the same subject is 0.500, and the alpha (significance) level 0.05 or 5%, a sample of 130 PLWHIV and FEP and 130 HIV negative with FEP will be required. In addition, this sample size accounts for and permits a 10% dropout rate over an anticipated 36-month accrual with a minimum follow-up of 12 months."

For the "Data management and analysis" section on p. 12, although some of the data can be analysed using a repeated measures ANOVA, consider using growth mixture modelling that is more robust.

Response: Thank you for the suggested rewording. This has been amended on page 9 and 11 respectively.

7. Provide some specifics of the distress plan under the "Potential risks" section on p. 13.

Response: The sites have specialist psychiatric services, both inpatient and outpatient, with a team available to attend to any adverse event that may arise during the interview process. Amended on page 12.

8. Consider adding and expanding upon the cultural component of this study; it's an important part of this protocol and has meaningful implications. You can potentially add a bit more to the 2nd to last paragraph in the Discussion on p. 15.

Response: Comment is noted and addressed as suggested in the discussion.

9. In Figure 1, add a brief description of each study objective as part of the "Objective 1" and "Objective 2" headings.

Response: Figure 1 has been amended accordingly.

Reviewer 2:

1. Are there any measures in place for the qualification of the diagnosis of FEP given the low mental health literacy and stigma in the study setting?

Response: The study is set at 5 academic clinical sites who all have experienced psychiatrists. The psychosis diagnosis is confirmed using the MINI schedule (Sheehan et al.) and where possible, collateral information is obtained from the family with the participants consent. Every participant's psychosis diagnosis is also correlated with the treating psychiatrist's assessment.

2. What are the plans for handling those who fail to stabilize after the 6 weeks of treatment? or the prevalent relapses associated with psychosis?

Response: Patients who are not able to consent within the 6 weeks of initiation of treatment are not included in this study but continue to receive treatment. Their data is collated in another first episode psychosis database study, which collates data on all patients with FEP irrespective of participation in the current study.

3. Spell the SOP training for the involved medical personnel to minimize incomplete data sets.

Response: All study personnel were trained before initiation of study and supported with annual refresher training on the different tools. PANSS training was done by an external trainer. Inter-rater assessments were also conducted. In addition, we held monthly investigator meetings to ensure adherence to data collection protocol and data is captured after recruitment so queries can be addressed. All investigators are connected and able to discuss concerns about tools digitally. When investigators had difficulty determining how to collect/capture data, the principal investigators, Dr Paruk and Prof Chiliza provided clinical guidance to the team.

4. Any acknowledgment of potential confounders given the multiplicity of healthcare pathways for mental illnesses in SSA? Also spell out if any rehabilitation treatments like occupational therapy are part of standard of care as they are evidenced to influence quality of life and other target outcomes.

Response: Pathways to care significantly influence access to care and hence also clinical features such as duration of untreated psychosis, age of presentation and symptom severity. Hence this study seeks to explore pathways to care using the WHO Encounter and includes several questions in the socio-demographic questionnaire relating to this aspect. We are keen to explore impact of pathways of care on sociodemographic and clinical variables. Rehabilitation programs are limited as many sites lack multi-disciplinary team to support this, however standard of care includes medication, supportive therapy, psychoeducation, referral to psychology if indicated, and management of comorbid substance use. Unfortunately access to OT/rehabilitation program is not routinely available in our limited resource setting and hence not included in text. We have included a sentence in limitations: This study does not include or measure psychosocial treatment and in future studies an intervention arm with psychosocial rehabilitation will provide useful information on efficacy of these treatments in a limited resource setting.

5. Is there a justification for only isiZulu translation given the range of local languages spoken in SA?

Response: The most widely spoken home/first language (>80%) among the almost 8 million inhabitants of KwaZulu-Natal is isiZulu. The reference is: Van der Merwe, I. & L. van Niekerk (1994). Language in South Africa: distribution and change. Stellenbosch: Department of Geography, Stellenbosch University.

6. What measures are in place for aiding attention and concentration as the participants take the 2hr cognitive assessment battery?

Response: Participants are offered breaks between assessments. Assessments are booked in the morning and participants are offered a drink break midway.

7. How often do you plan to test for HIV in the control group, and how will the positive cases be handled?

Response: HIV testing will be carried out at baseline as well as at each visit (3,6 and 12 months). A participant that tests positive on HIV ELISA test will be counselled and referred to the HIV clinic at the site for further management. This is added to text on page 10.

VERSION 2 – REVIEW

REVIEWER	NHUNZVI, CLEMENT University of Zimbabwe, Rehabilitation
REVIEW RETURNED	18-Apr-2021
GENERAL COMMENTS	Thank you for addressing my review comments. The paper read well and is clear. May need to state upfront how you handle possible variations of psychosis by aetiology e.g. HIV organic vs substance induced psychosis.